# FgAP1^σ^ Is Critical for Vegetative Growth, Conidiation, Virulence, and DON Biosynthesis in *Fusarium graminearum*

**DOI:** 10.3390/jof9020145

**Published:** 2023-01-21

**Authors:** Congxian Wu, Huilin Chen, Mingyue Yuan, Meiru Zhang, Yakubu Saddeeq Abubakar, Xin Chen, Haoming Zhong, Wenhui Zheng, Huawei Zheng, Jie Zhou

**Affiliations:** 1Fujian Universities Key Laboratory for Plant-Microbe Interaction, College of Life Sciences, Fujian Agriculture and Forestry University, Fuzhou 350002, China; 2Public Technology Service Center, Fujian Medical University, Fuzhou 350122, China; 3Department of Biochemistry, Faculty of Life Sciences, Ahmadu Bello University, Zaria 810107, Nigeria; 4State Key Laboratory of Ecological Pest Control for Fujian and Taiwan Crops, College of Plant Protection, Fujian Agriculture and Forestry University, Fuzhou 350002, China; 5Institute of Oceanography, College of Geography and Oceanography, Minjiang University, Fuzhou 350108, China

**Keywords:** *Fusarium graminearum*, wheat scab, AP1 complex, FgAP1^σ^

## Abstract

The AP1 complex is a highly conserved clathrin adaptor that plays important roles in regulating cargo protein sorting and intracellular vesicle trafficking in eukaryotes. However, the functions of the AP1 complex in the plant pathogenic fungi including the devastating wheat pathogen *Fusarium graminearum* are still unclear. In this study, we investigated the biological functions of FgAP1^σ^, a subunit of the AP1 complex in *F. graminearum*. Disruption of FgAP1^σ^ causes seriously impaired fungal vegetative growth, conidiogenesis, sexual development, pathogenesis, and deoxynivalenol (DON) production. The Δ*Fgap1^σ^* mutants were found to be less sensitive to KCl- and sorbitol-induced osmotic stresses but more sensitive to SDS-induced stress than the wild-type PH-1. Although the growth inhibition rate of the Δ*Fgap1^σ^* mutants was not significantly changed under calcofluor white (CFW) and Congo red (CR) stresses, the protoplasts released from Δ*Fgap1^σ^* hyphae were decreased compared with the wild-type PH-1, suggesting that FgAP1^σ^ is necessary for cell wall integrity and osmotic stresses in *F. graminearum*. Subcellular localization assays showed that FgAP1^σ^ was predominantly localized to endosomes and the Golgi apparatus. In addition, FgAP1^β^-GFP, FgAP1^γ^-GFP, and FgAP1^μ^-GFP also localize to the Golgi apparatus. FgAP1^β^ interacts with FgAP1^σ^, FgAP1^γ^, and FgAP1^μ^, while FgAP1^σ^ regulates the expression of *FgAP1^β^*, *FgAP1^γ^*, and *FgAP1^μ^* in *F. graminearum*. Furthermore, the loss of *FgAP1^σ^* blocks the transportation of the v-SNARE protein FgSnc1 from the Golgi to the plasma membrane and delays the internalization of FM4-64 dye into the vacuole. Taken together, our results demonstrate that FgAP1^σ^ plays vital roles in vegetative growth, conidiogenesis, sexual reproduction, DON production, pathogenicity, cell wall integrity, osmotic stress, exocytosis, and endocytosis in *F. graminearum*. These findings unveil the functions of the AP1 complex in filamentous fungi, most notably in *F. graminearum*, and lay solid foundations for effective prevention and control of Fusarium head blight (FHB).

## 1. Introduction

Clathrin-coated vesicles (CCVs) are among the major transport vesicles that contain two major structural components: clathrin and clathrin adaptor proteins (APs) [1,2]. Clathrin serves as a polymeric mechanical scaffold of these vesicles, and the recruitment of clathrin to membranes requires adaptors [3]. In eukaryotes, there are five distinct clathrin adaptor protein complexes, namely, AP1, AP2, AP3, AP4, and AP5, and they play important roles in vesicle assembly, recruitment of membrane cargoes and coat proteins, and vesicular transport [4,5]. Fungi possess only three functional AP complexes: AP1, AP2, and AP3; the AP4 and AP5 complexes have been lost through evolution [6].

Of the three AP complexes in fungi, the AP1 complex is the most conserved, and it is a heterotetrameric protein complex that binds membranes that are rich in phosphatidylinositol 4-phosphate, such as the membrane of the trans-Golgi network (TGN), and regulates cargo sorting between the TGN and recycling endosomes [7,8,9]. The AP1 complex consists of two large (∼100 kDa) subunits, beta-adaptin (β) and gamma-adaptin (γ), one medium-sized (∼50 kDa) subunit (μ), and one small (∼20 kDa) subunit sigma (σ) [1,10,11]. Its molecular architecture closely resembles that of AP2, the plasma-membrane-specific adaptor [12]. Previous studies have shown that the recruitment of the AP1 complex to membranes occurs following physical interaction with phosphatidyl-inositol-4-phosphate (PI4P), the HEATR5 protein Laa1 and its cofactor Laa2, and the small GTPase Arf1 [13,14,15]. Cargo recruitment is mediated by the AP1μ subunit which directly binds tyrosine motifs on the cargo proteins [16]. The ears and linkers of the β subunit specifically bind to clathrin and clathrin-accessory proteins, though the mammalian β subunit is also found to bind the dileucine motif of cargo proteins [17,18]. The σ subunit is involved in cargo selection also by binding dileucine motifs [19,20].

AP1 localizes to the TGN, and the TGN localization of this protein complex depends on the small GTPase, Arf1, and the phosphoinositide, PI-P [12]. AP1 is also required for the polarized localization of many plasma membrane proteins [21]. The AP1 protein complex participates in both clathrin-dependent and clathrin-independent processes in *Saccharomyces cerevisiae* [22]. The loss of the AP1 complex subunits *β, γ, μ*, and *σ* in *Schizosaccharomyces pombe* individually showed distinct phenotypes including growth sensitivity to drugs or temperature [23]. In *Aspergillus nidulans*, the AP1 complex is required for the release of the secretory vesicle, polar sorting, endosome recycling, and cytoskeleton organization [24]. AP1 functions as a heterotetrameric complex in *Toxoplasma gondii*, and deletion of the *AP1μ* subunit impaired the sorting of rhoptry and microneme proteins, parasite division, and host infection [25]. In *Arabidopsis thaliana*, AP1 localizes to trans-Golgi network (TGN) and regulates trafficking from the TGN to the vacuole and plasma membrane (exocytosis) [26,27]. Further study found that AP1σ (AP1σ1 and AP1σ2) is crucial for pollen wall formation by mediating the secretion of lipid transfer proteins (LTPs) and sporopollenin synthase acyl-CoA synthetase 5 (ACOS5) from tapetum into the anther locule [28]. Disruption of various subunits (γ1, β1, μ1, or σ1) of the AP1 complex in mice causes prenatal lethality and severe growth defects [29,30,31,32,33]. There are three genes encoding the AP1 small subunit σ in humans, and disruption of *AP1S1* (σ1A) impaired the development of the skin and spinal cord and caused reduced pigmentation and severe motility deficits [34]. AP1B is required for the polarized distribution of many membrane proteins to the basolateral surface of LLC-PK1 kidney cells [35]. Mutations in *AP1S3* (σ1C) cause a severe autoinflammatory skin disorder (called pustular psoriasis) due to defects in vesicular trafficking, and *AP1S3* (σ1C) is required for Toll-like receptor homeostasis [36]. However, the role of the AP1 complex in plant pathogenic fungi is still unclear.

*Fusarium graminearum* is a filamentous fungus that causes an economically devastating disease called Fusarium head blight (FHB) in wheat and other cereals, and it has become one of the most serious problems for agricultural production worldwide [37,38,39,40]. In addition to enormous losses in wheat yield, mycotoxins such as deoxynivalenol (DON), nivalenol (NIV), and zearalenone (ZEA) produced by the fungus in infected grains impose serious threats to food security and human health [41,42]. Recent studies demonstrated that intracellular vesicle trafficking regulates fungal development, pathogenicity, and DON (mycotoxin) production in *F. graminearum* [43,44,45,46,47,48].

Although the roles of the AP1 complex in *S. cerevisiae* and mammals have been studied, the functions of this complex in plant pathogenic fungi are still unknown. Our previous studies revealed that the well-conserved FgAP2 complex subunits FgAP2^β^, FgAP2^α^, FgAP2^mu^, and FgAP2^σ^ are essential for the polarized growth, development, and pathogenicity of *F. graminearum* and regulate the polar localization of the lipid flippases FgDnfA and FgDnfB [49]. In this study, we functionally characterized the AP1 complex subunit FgAP1^σ^ in the development and pathogenesis of *F. graminearum* using molecular genetic techniques. Our results showed that FgAP1^σ^ is essential for development, cell wall integrity, the response to osmotic stress, DON production, pathogenicity, and the transport of the v-SNARE FgSnc1 from the Golgi to the plasma membrane, and it causes the delay of the endocytosis process in *F. graminearum*. Moreover, FgAP1^σ^ is localized to endosomes and the Golgi apparatus and regulates the expression of *FgAP1^β^*, *FgAP1^γ^*, and *FgAP1^μ^* in *F. graminearum*. The findings improve our understanding of the functions of the AP1 complex as well as its involvement in host–pathogen interaction and lay the foundation for effective prevention and control of FHB.

## 2. Materials and Methods

### 2.1. Fungal Strains, Media, and Culture Conditions

The wild-type PH-1 and all transformants used in this study (Appendix A) were cultured at 28 °C on starch yeast media (SYM: 0.2% yeast extract, 1% starch, 0.3% sucrose, 2% agar). Vegetative growth of the various strains were assayed on starch yeast media (SYM), solid complete media (CM: 0.6% casein hydrolyzate, 1% sucrose, 0.6% yeast extract, and 2% agar), and minimal media (MM: 0.52 g KCl, 6 g NaNO_3_, 1.52 g KH_2_PO_4_, 10 g glucose, 0.52 g MgSO_4_, 0.1% trace elements (*v/v*), and 2% agar) at 28 °C for 3 days. Solid CM medium was supplemented with different inhibitors to test the response of the strains to different stresses. For conidiation assay, the various strains were cultivated in liquid carboxymethylcellulose (CMC) medium, cultured for 3 days to induce conidia production as described previously [50]. Conidia from PH-1 and all the mutants were visualized using an Olympus BX51 microscope. Perithecia formation assay was conducted on carrot agar medium as previously reported [51]. The vegetative growth, conidiation, and perithecia formation assays were repeated three times.

### 2.2. Gene Deletion and Complementation 

We searched for the predicted *F. graminearum* AP1 complex proteins using S. cerevisiae AP1 complex proteins (Aps1p, Apm1p, Apl2p, and Apl4p) as a reference to perform a BLAST search against the available fungal genome and identified FGSG_10034, FGSG_17141, FGSG_06317, and FGSG_01893 genetic loci encoding the homologues of AP1 complex proteins in *F. graminearum*. For convenience, we named these genes FgAP1^σ^, FgAP1^u^, FgAP1^β^, and FgAP1^γ^, respectively.

*F. graminearum* protoplast preparation and fungal transformation were conducted as described previously [52]. Deletions of *FgAP1^σ^* (FGSG_10034) genes were achieved using a split-marker approach [53]. Appendix A presents all the primers used for gene deletion. The FgAP1^σ^ knockout mutants were successfully generated and further verified by Southern blot. For complementation assay, green fluorescent protein (GFP) fragment was tagged at the C-terminus of the *FgAP1^σ^* gene sequence. We amplified the *FgAP1^σ^* coding sequence under the control of its native promoter. The GFP fragment that was fused with the *FgAP1^σ^* gene sequence was inserted into a pKNT plasmid to generate FgAP1^σ^-GFP vector using the Cloning Kit (Vazyme Biotech Co., Ltd., Nanjing, China). The resulting FgAP1^σ^-GFP construct was amplified using specific primer pairs and sequenced to verify its successful insertion into the plasmid. Finally, the FgAP1^σ^-GFP vector was transformed into the Δ*Fgap1^σ^* mutant protoplasts to generate the complemented strain which was phenotypically similar to the PH-1.

### 2.3. Construction of Fusion Vectors

The native promoters and coding sequences of FgAP1^σ^-GFP, FgAP1^β^-GFP, FgAP1^γ^-GFP, and FgAP1^μ^-GFP constructs, as well as the GFP fragment (isolated from the FgGdt1-GFP vector) [51] were amplified using specific primer pairs (Appendix A). The GFP fragment was tagged at the C-terminus of FgAP1^σ^, FgAP1^β^, FgAP1^γ^, and FgAP1^μ^ and cloned into pKNT vector. The resulting FgAP1^σ^-GFP, FgAP1^β^-GFP, FgAP1^γ^-GFP, and FgAP1^μ^-GFP constructs were amplified by PCR using specific primer pairs (Appendix A) and confirmed by sequencing to verify successful insertion. FgKex2 was tagged with mCherry at its C-terminus, while FgSnc1 was tagged with GFP at its N-terminus, under the control of their respective native promoters, to generate FgKex2-mCherry and GFP-FgSnc1 vectors, respectively. The constructed vectors were transformed into the wild-type PH-1 or Δ*Fgap1^σ^* mutant protoplasts.

### 2.4. RNA Extraction and Quantitative Reverse Transcription PCR

Total RNA was extracted from mycelia of the wild type and mutants cultivated in liquid CM and incubated at 28 °C under constant shaking at 110 rpm for 2 days, or in liquid trichothecene biosynthesis induction (TBI) media and incubated at 28 °C in the dark for 3 days, using an Eastep Super Total RNA Extraction Kit (Promega, Shanghai, China ). Reverse transcription was performed to generate complementary DNA (cDNA) using a Reverse Transcription Kit (Takara, Tokyo, Japan). The expression level of each gene was quantified by qRT-PCR using a TB GREEN kit (Takara, Tokyo, Japan) with specific primers (Appendix A). The *Fusarium graminearum* β-tubulin gene was used as an endogenous reference gene. Relative gene expressions were finally calculated using the 2^−ΔΔCT^ method according to [54]. All qRT-PCR assays were repeated three times.

### 2.5. Subcellular Localization Assay

For the observation of *F. graminearum* mycelial morphology and cellular localization, the various strains were grown in liquid or on solid CM media at 28 °C for 24 h, and microscopic visualizations were performed using a Nikon A1R laser scanning confocal microscope (Nikon, Tokyo, Japan). Conidia from the wild-type PH-1 and Δ*Fgap1^σ^* strains were stained with the cell-wall-damaging agent calcofluor white (CFW) at a final concentration of 10 μg/mL for visualization of cell walls and septa using an Olympus BX51 microscope (Olympus, Tokyo, Japan). Similarly, hyphae from the wild-type PH-1 and Δ*Fgap1^σ^* strains were stained with the fluorescent dye FM4-64 at a final concentration of 4 µM, incubated, and visualized for observation of endocytosis using the Nikon A1R laser scanning confocal microscope (Nikon, Tokyo, Japan).

### 2.6. Yeast Two-Hybrid

For yeast two-hybrid (Y2H) assays, the full length cDNAs of FgAP1^σ^, FgAP1^β^, FgAP1^γ^, and FgAP1^μ^ were amplified by PCR using specific primers (Appendix A). The FgAP1^σ^, FgAP1^β^, and FgAP1^μ^ fragments were cloned into pGBKT7 vectors pre-digested with *Nde* I and *Eco*R I as bait vectors to generate FgAP1^σ^-BD, FgAP1^β^-BD, and FgAP1^μ^-BD recombinant vectors, respectively. In addition, FgAP1^β^, FgAP1^μ^, and FgAP1^γ^ fragments were cloned into pGADT7 pre-digested with *Nde* I and *Eco*R I as prey vectors to generate FgAP1^β^-AD, FgAP1^μ^-AD, and FgAP1^γ^-AD recombinant vectors, respectively. All prey and bait vectors were confirmed by sequencing and cotransformed into *S. cerevisiae* AH109 strain according to a previous study [55]. Interaction between pGBKT7-53 and pGADT7 served as positive control, while that between pGBKT7-Lam and pGADT7 served as negative control.

### 2.7. Pathogenicity and DON Production Assays

The pathogenicity of the various strains on flowering wheat heads was conducted as described previously [51]. For wheat spikelet infection assay, the wild-type PH-1 and Δ*Fgap1^σ^* strains were inoculated in liquid CMC medium, and their conidia concentrations were adjusted to 4 × 10^4^ cells/mL. Conidia suspensions were inoculated to fresh wheat coleoptile wounds, and disease symptoms were observed 7 days after incubation at 25 °C. For DON production assay, mycelia plugs from the strains were cultured in liquid trichothecene biosynthesis induction (TBI) media and incubated at 28 °C in the dark and without shaking for 7 days. An ELISA-based DON detection kit (FINDE, Shenzhen, China) [51] was used to measure DON levels in the liquids while the mycelia were collected, dried, and weighed. The DON production was standardized as 1 g dry weight of mycelia. The experiment was repeated three times.

## 3. Results

### 3.1. FgAP1^σ^ Gene Deletion and Confirmation

Bioinformatics analysis showed that the *F. graminearum* AP1 complex contains four subunits: FgAP1^σ^, FgAP1^u^, FgAP1^β^, and FgAP1^γ^. To understand the biological functions of the FgAP1 complex in *F. graminearum*, a targeted gene replacement strategy was used to delete FgAP1 complex subunits in PH-1 (wild type). Of the four FgAP1 genes, only FgAP1^σ^ was deleted successfully, while FgAP1^u^, FgAP1^β^, and FgAP1^γ^ knockout mutants could not be obtained despite multiple deletions, suggesting that they are essential for the survival of *F. graminearum*. FgAP1^σ^ encodes a 181-amino-acid protein that shares 56% identity with *S. cerevisiae* Aps1. Phylogenetic analysis and sequence alignment of FgAP1^σ^ and other AP1^σ^ proteins showed that FgAP1^σ^ is highly conserved in eukaryotes, especially in the filamentous fungi Fusarium oxysporum, Fusarium verticillioides, Manaporthe oryzae, and Neurospora crassa (Figure 1A,B). The ΔFgap1^σ^-14 and ΔFgap1^σ^-19 mutants were further confirmed by Southern blot analysis, which showed a 3.95 kb band in the FgAP1^σ^ deletion mutants in contrast to a 5.09 kb band in the wild-type PH-1 (Appendix A). In addition, FgAP1^σ^ was tagged with GFP at its C-terminus, under the control of its native promoter to form the FgAP1^σ^-GFP vector. The constructed vectors were transformed into the ΔFgap1^σ^-14 mutant protoplasts to generate the complemented strain ΔFgap1^σ^-C.

### 3.2. FgAP1^σ^ Plays an Important Role in Vegetative Growth of F. graminearum

To determine whether FgAP1^σ^ is required for the vegetative growth and colony morphology of *F. graminearum*, the wild-type PH-1, FgAP1^σ^ deletion mutants (ΔFgap1^σ^-14, ΔFgap1^σ^-19), and the complementation strain ΔFgap1^σ^-C were cultured on CM, SYM, and MM plates for 3 days. Compared with the PH-1 and ΔFgap1^σ^-C strains, the ΔFgap1^σ^-14 and ΔFgap1^σ^-19 mutants showed significantly reduced vegetative and aerial hyphal growths (Figure 2A,B). Close microscopic examinations showed increased mycelial branching in the ΔFgap1^σ^-14 and ΔFgap1^σ^-19 mutants compared with the PH-1 and ΔFgap1^σ^-C strains. In addition, the morphology of the mutants appeared irregular (Figure 2C). These results suggest that FgAP1^σ^ is important for *F. graminearum* vegetative and polarized growth.

### 3.3. FgAP1^σ^ Is Required for Conidiation and Sexual Development of F. graminearum

Asexual conidia and sexual ascospores produced by *F. graminearum* function as the important inocula that infect flowering wheat heads. We investigated the asexual reproduction of ΔFgap1^σ^ mutants and found that the conidiation of the ΔFgap1^σ^-14 and ΔFgap1^σ^-19 mutants were significantly decreased compared with those of PH-1 and ΔFgap1^σ^-C strains (Figure 3A). Moreover, microscopic examinations of the conidia indicated that 95.45% of the conidia from the ΔFgap1^σ^-14 mutant and 99.41% of the conidia from the ΔFgap1^σ^-19 mutant had one or no septum, unlike most of the PH-1 and complemented strain which possessed at least three septa per conidium (Figure 3B,C). Although the conidia morphology of the ΔFgap1^σ^-14 and ΔFgap1^σ^-19 mutants was obviously abnormal, their germinations on CM media after 4 h or 8 h were not affected compared with the PH-1 and ΔFgap1^σ^-C strains (Figure 3D). To investigate the roles of FgAP1^σ^ in the fungal sexual reproduction, the wild-type PH-1, ΔFgap1^σ^-14, ΔFgap1^σ^-19, and ΔFgap1^σ^-C strains were grown on carrot agar for perithecia and ascospore formation. As shown in Figure 3E, the loss of FgAP1^σ^ in *F. graminearum* completely abolished perithecia and ascospore formation. Our results suggest that FgAP1^σ^ plays important roles in reproduction in *F. graminearum*.

### 3.4. FgAP1^σ^ Is Critical for F. graminearum Virulence and DON Production

To investigate the roles of FgAP1^σ^ in the pathogenicity of *F. graminearum* to its host plants, we conducted infection assays on flowering wheat heads and wheat coleoptiles. Mycelia plugs from the wild-type PH-1, ΔFgap1^σ^-14, ΔFgap1^σ^-19, and ΔFgap1^σ^-C strains were inoculated on flowering wheat heads under moist conditions for two weeks. The wild-type PH-1 and ΔFgap1^σ^-C strains caused severe head blight symptoms, while the ΔFgap1^σ^-14 and ΔFgap1^σ^-19 mutants showed significantly decreased infection capability and did not produce obvious symptoms (Figure 4A). We further conducted virulence assays of PH-1 and the ΔFgap1^σ^ mutants on young wheat coleoptiles and obtained similar results for each strain (Figure 4B). DON is well characterized as an important mycotoxin and virulence factor in the pathogenicity of *F. graminearum* on wheat. To investigate whether FgAP1^σ^ is required for DON production, we cultured the PH-1, ΔFgap1^σ^, and ΔFgap1^σ^-C strains in liquid TBI media at 28 °C for 7 days under dark conditions and checked the levels of DON production afterwards. We found that DON production was significantly decreased in the ΔFgap1^σ^-14 and ΔFgap1^σ^-19 mutants compared with the wild-type PH-1 and ΔFgap1^σ^-C strains (Figure 4C), which is consistent with our pathogenicity tests. Next, we examined the expression levels of the DON-biosynthesis-related genes FgTRI6, FgTRI10, FgTRI1, FgTRI4, FgTRI5, and FgTRI12 in the ΔFgap1^σ^ mutants harvested from TBI media by quantitative reverse transcription PCR (qRT-PCR) analysis. Our results showed that the transcription levels of the various genes were significantly downregulated in the ΔFgap1^σ^-14 and ΔFgap1^σ^-19 mutants compared with those in the wild-type PH-1 (Figure 4D). Collectively, these results demonstrate that FgAP1^σ^ regulates virulence and DON biosynthesis in *F. graminearum*.

### 3.5. FgAP1^σ^ Is Important for Cell Wall Integrity and Response to Osmotic Stress in F. graminearum

To examine the impact of FgAP1^σ^ deletion on the responses of *F. graminearum* to various types of stresses, we cultured the various strains on CM media containing the cell-wall-perturbing agents Congo red (CR) and calcofluor white (CFW), the cell membrane inhibitor SDS, and the osmotic-stress-inducing agents KCl and sorbitol for 3 days and monitored growth inhibition. The results obtained from these bioassays showed that the growth inhibition rates of the ΔFgap1^σ^-14 and ΔFgap1σ-19 mutants under SDS-induced stress were significantly higher than in the presence of other agents, suggesting that ΔFgap1^σ^ mutants are more sensitive to SDS-induced stress (Figure 5A–C). In addition, the growth inhibition rates of the ΔFgap1^σ^ mutants were not significantly changed under CFW and CR stresses compared with PH-1 and ΔFgap1^σ^-C strains, which means that they are not sensitive to CFW and CR stresses (Figure 5A–C). Next, we examined the release of protoplasts by the ΔFgap1^σ^ mutants in the presence of cell-wall-degrading enzymes to further confirm the integrity of the cell walls of the ΔFgap1^σ^ mutants. Our results showed that the hyphae of the ΔFgap1^σ^-14 and ΔFgap1^σ^-19 mutants released less protoplasts than those of the wild-type PH-1 after 2 h and 3 h of incubation with cell-wall-degrading enzymes (Figure 5D), indicating that the loss of FgAP1^σ^ affects the fungal cell wall integrity. Further study found that the ΔFgap1^σ^-14 and ΔFgap1^σ^-19 mutants were more tolerant to osmotic stress on CM media supplemented with the osmotic stress agents KCl and sorbitol than PH-1 (Figure 5A–C), indicating that the ΔFgap1^σ^ mutants are less sensitive to KCl- and sorbitol-induced osmotic stresses. 

### 3.6. FgAP1^σ^ Localizes to Endosomes and Golgi Apparatus

We further investigated the subcellular localization of the FgAP1 complex and found that FgAP1^σ^-GFP partially colocalized with FM4-64-positive endosomes in *F. graminearum* (Figure 6). In yeast, AP1^σ^ localizes to the Golgi [4]. To check whether FgAP1^σ^-GFP is localized to the Golgi apparatus in addition to its endosome localization, we cotransformed the FgAP1^σ^-GFP vectors with another vector expressing the Golgi marker FgKex2-mCherry [51] into the protoplasts of PH-1. As shown in Figure 6, FgAP1^σ^-GFP also localized to the Golgi apparatus. To gain insight into the subcellular localization of other subunits of the FgAP1 complex, we constructed the FgAP1^β^-GFP, FgAP1^γ^-GFP, and FgAP1^μ^-GFP vectors and cotransformed them with the FgKex2-mCherry vector into the PH-1 protoplasts. Our results showed that FgAP1^β^-GFP, FgAP1^γ^-GFP, and FgAP1^μ^-GFP all colocalized with FgKex2-mCherry at the Golgi apparatus (Figure 6), suggesting that the FgAP1 complex localizes to the Golgi apparatus in *F. graminearum*.

### 3.7. The Relationship of FgAP1^σ^ with Other Subunits of AP1 Complex 

To investigate the relationship of the various subunits in the AP1 complex, we conducted yeast two-hybrid assays and found that FgAP1^β^ interacts with FgAP1^σ^, FgAP1^γ^, and FgAP1^μ^, but we could not find any positive interaction between FgAP1^σ^ and FgAP1^γ^ nor between the former and FgAP1^μ^ (Figure 7A). Therefore, a proposed model depicting the relationship between the AP1 complex subunits is depicted in Figure 7B, which suggests that FgAP1^β^, FgAP1^σ^, FgAP1^γ^, and FgAP1^μ^ function as a complex in *F. graminearum*.

To further investigate the regulatory role of FgAP1^σ^ with the other three subunits, we tested the transcription levels of FgAP1^β^, FgAP1^γ^, and FgAP1^μ^ in the ΔFgap1^σ^ mutants by qRT-PCR and found that they are all downregulated in the ΔFgap1^σ^-14 and ΔFgap1^σ^-19 mutants compared with the wild-type PH-1 (Figure 7C). To examine whether FgAP1^σ^ regulates the subcellular localization of FgAP1^β^, FgAP1^γ^, and FgAP1^μ^, we transformed FgAP1^β^-GFP, FgAP1^γ^-GFP, and FgAP1^μ^-GFP vectors into the ΔFgap1^σ^ protoplasts and checked the localizations of their protein products in the absence of FgAP1^σ^. Our results showed that the punctate localizations of FgAP1^β^-GFP, FgAP1^γ^-GFP, and FgAP1^μ^-GFP were not affected in the ΔFgap1^σ^ mutant (Figure 7D). 

### 3.8. FgAP1^σ^ Is Required for Plasma Membrane Localization of the v-SNARE FgSnc1

Previous studies have shown that FgSnc1 regulates the polarized secretion and fusion of vesicles in *F. graminearum*. To understand the roles of FgAP1^σ^ in vesicle transport processes, we constructed the GFP-FgSnc1 vector and transformed it into the wild-type PH-1 and ΔFgap1^σ^ protoplasts and subsequently observed the cellular localizations of GFP-FgSnc1 by confocal microscopy. In PH-1, GFP-FgSnc1 was observed to be clearly localized to the plasma membrane, and it also concentrated at the Spitzenkörper (SPK) of growing hyphal cells in PH-1 (Figure 8A). However, in the ΔFgap1^σ^ mutant expressing GFP-FgSnc1, the hyphal tip localization of this protein was still present but not its plasma membrane localization (Figure 8A). These results suggest that FgAP1^σ^ regulates the secretion of FgSnc1 from the Golgi to the plasma membrane.

### 3.9. Loss of FgAP1^σ^ Results in Delayed Internalization of FM4-64 into Vacuole Membrane

To investigate whether FgAP1^σ^ is involved in endocytosis in *F. graminearum*, we monitored the endocytic uptake of the fluorescent dye FM4-64 by the wild type and mutants at different time points. As shown in Figure 8B, the FM4-64 signal was detected localized on the plasma membrane of all the strains within 1 min of staining. After 15 min, endosomes (FM4-64-labeled vesicles) appeared inside the wild-type PH-1 and the complemented strain ΔFgap1^σ^-C, indicating normal endocytosis. However, a large amount of FM4-64 remained in the membrane of the ΔFgap1^σ^-14 and ΔFgap1^σ^-19 mutants. At 45 min after staining, FM4-64 was found to internalize into the vacuole membrane through the endosomes of hyphal cells in the wild-type PH-1 and ΔFgap1^σ^ mutants. These results collectively reveal that FgAP1^σ^ delays the internalization of FM4-64 into the vacuole membrane in *F. graminearum*.

## 4. Discussion

The AP1 complex is a conserved heterotetrameric protein complex that consists of AP1^σ^, AP1^β^, AP1^γ^, and AP1^μ^ subunits in eukaryotes [8,56,57]. Previous studies in yeast, *T. gondii*, *A. thaliana*, and mammals demonstrated that the AP1 complex plays fundamental roles in vesicle assembly, cargo sorting, and intracellular vesicular transport [22,25,33,36,58]. In this study, we identified four candidate subunits (FgAP1^σ^, FgAP1^β^, FgAP1^γ^, and FgAP1^μ^) of the AP1 complex in *F. graminearum* and then investigated the biological functions of the small subunit FgAP1^σ^ as well as the relationship of FgAP1^σ^ with the other subunits in the plant fungal pathogen *F. graminearum*. Our results showed that FgAP1^σ^ localizes to endosomes and the trans-Golgi network (TGN) and is critical for fungal development and pathogenicity to wheat. Further study found that FgAP1^σ^ plays an important role in cell wall integrity and response to osmotic stresses. It also regulates the transcription levels of other AP1 subunits as well as the secretory process from the Golgi apparatus to the plasma membrane and delays the endocytosis process. 

In *S. pombe*, Aps1p (σ) directly interacts with Apl4p (γ), while Apm1p (μ) directly interacts with Apl2p (β1). The localization of Apl4p (γ), Apm1p (μ), and Apl2p (β1) were significantly changed in the Δ*aps1* (Δ*ap1^σ^*) mutant [23]. However, the relationships between the various subunits of the AP1 complex in plant pathogenic fungi are still unclear. In this study, yeast two-hybrid assays showed that FgAP1^β^ interacts with FgAP1^σ^, FgAP1^γ^, and FgAP1^μ^ and form the AP1 complex in *F. graminearum*. However, the punctate localizations of FgAP1^γ^, FgAP1^β^, and FgAP1^μ^ remain unchanged in the Δ*Fgap1^σ^* mutant in *F. graminearum* (Figure 7A,B,D). This clearly demonstrates a difference in the interaction of the AP1 complex subunits between yeast and plant pathogenic fungi. In addition, FgAP1^σ^ regulates the transcription levels of *FgAP1^β^*, *FgAP1^γ^*, and *FgAP1^μ^* (Figure 7C), suggesting that the loss of a single component of the FgAP1 complex affects the homeostasis of other components in *F. graminearum*. Furthermore, the loss of FgAP1^σ^ showed clear pleiotropic phenotypes, including defects in vegetative growth, conidia production, conidia morphology, sexual reproduction, and pathogenicity. We were able to delete *FgAP1^σ^* but not the genes encoding the other subunits of the FgAP1 complex (*FgAP1^β^*, *FgAP1^γ^*, and *FgAP1^μ^*) in PH-1 (wild type), suggesting that *FgAP1^β^*, *FgAP1^γ^*, and *FgAP1^μ^* are indispensable for the survival of *F. graminearum*. The loss of AP1 subunits in *S. pombe*, *T. gondii*, *A. thaliana*, and humans results in serious phenotypic defects [23,25,26,27,28,34,35,36]. Moreover, disruption of the *AP1^mu1A^* or *AP1^γ^* genes in mice resulted in embryonic lethality [31,32]. These results suggest that the AP1 complex plays important roles in normal growth and development in eukaryotic organisms.

Fungal cell walls have a complex structure and can be remodeled by many stresses, developmental processes, and plant infection [59,60]. The endocytic cargo adaptor complex 2 (FgAP2) is required for cell wall integrity by interacting with the sensor FgWsc2B in *F. graminearum* [61]. In the present study, although the Δ*Fgap1^σ^-14* and Δ*Fgap1^σ^-19* mutants were not sensitive to CFW and CR stresses, the number of protoplasts released by the mutants was very low (Figure 5A–D). Moreover, the mycelial growth of the Δ*Fgap1^σ^* mutants was promoted in the presence of KCl and sorbitol (Figure 5A–C). These results indicate that FgAP1^σ^ positively regulates cell wall integrity but negatively regulates osmotic stress resistance in *F. graminearum*.

AP1B is required for the polarized distribution of many membrane proteins to the basolateral surface of LLC-PK1 kidney cells [35]. Polarized growth is fundamental for fungal optimal survival and infection [62]. Our results showed that FgAP1^σ^ is required for polarized growth in *F. graminearum* (Figure 2C), suggesting a conserved role of the FgAP1 complex in polarized growth. AP1 binds membranes that are rich in phosphatidylinositol 4-phosphate, such as the trans-Golgi network (TGN) membrane, while AP2 associates with phosphatidylinositol 4,5-bisphosphate of the plasma membrane [9]. The FgAP2 complex colocalizes with actin patch components and regulates polarized growth in *F. graminearum* as shown by a previous study [49]. Here, we found that the FgAP1 complex, like FgAP2, localizes to TGN and is required for polarized growth in *F. graminearum* [49].

Previous studies have shown that the AP1 complex regulates cargo sorting within the TGN, endosomes, lysosomes, and plasma membrane [7,8,14,58]. In yeast, the clathrin adaptor protein complex 1 (AP1) is required for recycling of Chs3p and Tlg1p from the early endosome to the TGN [63]. *A. thaliana* AP1 localizes to the TGN and regulates trafficking to the vacuole and to the plasma membrane (exocytosis) [26,64,65]. In *A. nidulans*, AP1 is involved in anterograde sorting of both Rab11-labeled secretory vesicles and Rab5-dependent endosome recycling, as well as cytoskeleton-dependent polarized cargo traffic [24]. Soluble N-ethylmaleimide-sensitive factor (NSF) attachment protein receptors (SNAREs) are sorted into clathrin-coated vesicles by direct interaction with clathrin adaptors [66]. FgSnc1 is a SNARE protein; it localizes to the Spitzenkörper (SPK) and plasma membrane and regulates the fusion of vesicles and polarized secretion in *F. graminearum* [67,68]. Snc1 mediates the fusion of vesicles from the Golgi with the plasma membrane in yeast [69]. Our results showed that the fluorescence signal of GFP-FgSnc1 could not be detected at the plasma membrane of the Δ*Fgap1^σ^* mutant (Figure 8A), suggesting that FgAP1^σ^ is involved in polarized trafficking of FgSnc1 from the Golgi to the plasma membrane and is required for the pathogenicity of *F. graminearum*, consistent with previous studies which demonstrated that FgMsb3, FgGyp1, and FgRab1 mediate polarized trafficking and are important for plant infection in *F. graminearum* [43,44,68]. Endocytosis is essential for transporting extracellular or membrane-localized macromolecules to the cytoplasm which ensures that nutrients reach the cell [5]. The AP2 complex has been demonstrated to function in endocytosis in various species [58,70,71,72,73], while the role of the AP1 complex in endocytosis is obscure. We found that the endocytic uptake of FM4-64 was delayed in the Δ*Fgap1^σ^* mutants (Figure 8B), suggesting that FgAP1^σ^ may be involved in the endocytosis process. 

DON mycotoxin is known as an important virulence factor during infection of *F. graminearum* to wheat [74,75,76]. DON production in the Δ*Fgap1^σ^* mutants was found to decrease significantly compared with PH-1 (Figure 4C). The development of full disease symptoms was also perturbed in the Δ*Fgap1^σ^* mutants, probably due to the significant reduction in DON production. In summary, we demonstrated for the first time that FgAP1^σ^ is indispensable for vegetative growth, conidiation, conidia morphology, sexual reproduction, pathogenicity, and DON biosynthesis in *F. graminearum*. Moreover, FgAP1^σ^ is critical for the regulation of cell wall integrity, response to osmotic stresses, and vesicle fusion with target membranes. Therefore, our results showed that FgAP1^σ^ is critical for growth and development, virulence, and DON biosynthesis in *F. graminearum*. Since the expression levels of the *FgTRI* genes were affected in the Δ*Fgap1^σ^* mutant, then reduce DON production, further studies should examine whether the secretion of DON mycotoxin in the mutant is affected and investigate the mechanism of protein secretion as well as screening of FgAP1^σ^ cargos so as to unveil whether the observed phenotypic defects of the Δ*Fgap1^σ^* mutant are a direct result of the disruption of FgAP1^σ^ function in vesicle transport and cargo sorting in *F. graminearum*. 

## Figures and Tables

**Figure 1 jof-09-00145-f001:**
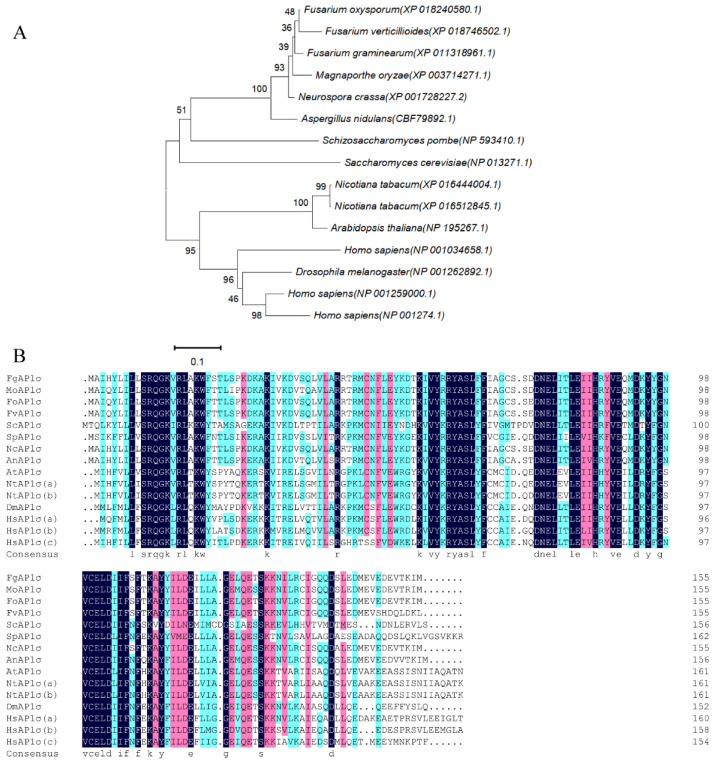
Bioinformatics analyses of FgAP1^σ^ homologues. (**A**) Phylogenetic relationship of FgAP1^σ^ protein from different organisms. The gene names are from the Broad Institute genome annotation: *Fusarium graminearum* (XP_011318961.1), Magnaporthe oryzae (XP_003714271.1), Fusarium oxysporum (XP_018240580.1), Fusarium verticillioides (XP_018746502.1), Saccharomyces cerevisiae (NP_013271.1), Schizosaccharomyces pombe (NP_593410.1), Neurospora crassa (XP_001728227.2), Aspergillus nidulans (CBF79892.1), Arabidopsis thaliana (NP_195267.1), Nicotiana tabacum (XP_016444004.1), Nicotiana tabacum (XP_016512845.1), Homo sapiens (NP_001259000.1), Homo sapiens (NP_001274.1), Homo sapiens (NP_001034658.1), Drosophila melanogaster (NP_001262892.1). (**B**) Multiple alignment of amino acid sequences of FgAP1^σ^ homologues.

**Figure 2 jof-09-00145-f002:**
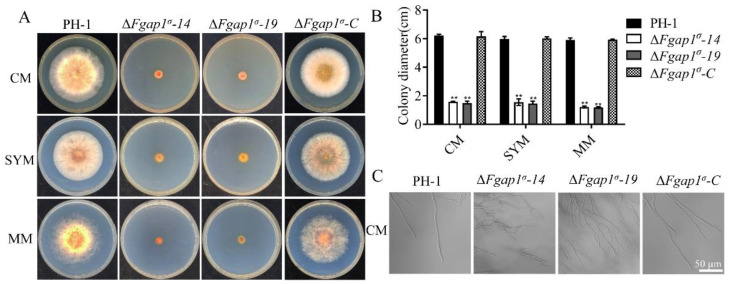
FgAP1^σ^ is required for the vegetative growth of *F. graminearum*. (**A**) Vegetative growth of the wild-type PH-1, ΔFgap1^σ^ mutants, and the complemented strain ΔFgap1^σ^-C after growth on complete media (CM), starch yeast medium (SYM), and minimal medium (MM) agar at 28 °C for 3 days. (**B**) Statistical analysis of the colony diameters of the PH-1, ΔFgap1^σ^, and ΔFgap1^σ^-C strains on CM, SYM, and MM media after 3 days. Level of significance was measured using an unpaired *t*-test (** *p* < 0.01). (**C**) Mycelial morphologies of PH-1, ΔFgap1^σ^, and ΔFgap1^σ^-C strains on CM medium. Hyperbranching was observed in the ΔFgap1^σ^ mutants. Bar = 50 μm.

**Figure 3 jof-09-00145-f003:**
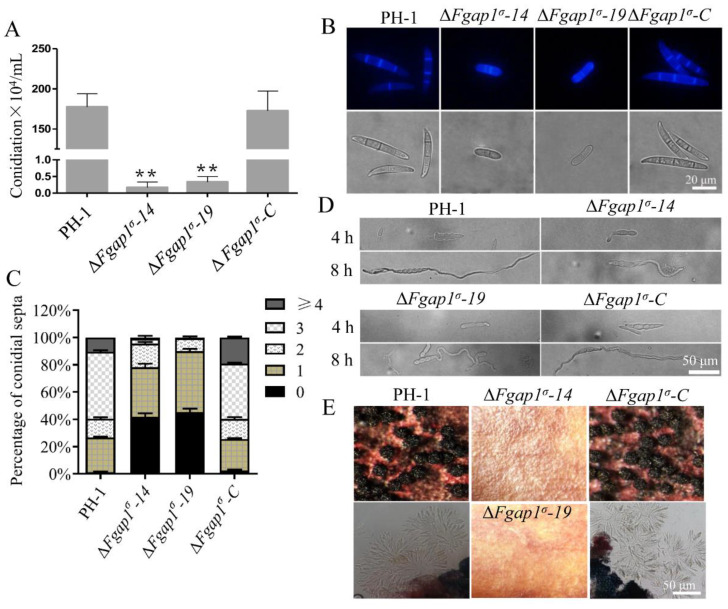
FgAP1^σ^ is involved in the conidiation and sexual development of *F. graminearum*. (**A**) Conidiation of the wild-type PH-1, ΔFgap1^σ^, and ΔFgap1^σ^-C strains in liquid CMC media for 3 days. Level of significance was measured using an unpaired *t*-test (** *p* < 0.01). (**B**) Conidial morphology was observed under a light microscope after the PH-1, ΔFgap1^σ^, and ΔFgap1^σ^-C strains were cultured in liquid CMC media for 3 days. Bar = 20 μm. (**C**) The number of septa in the conidia produced by the indicated strains. (**D**) Germination of the indicated strains in CM media after 4 h and 8 h of inoculation. Bar = 50 μm. (**E**) Images of the perithecia and ascospores produced by the PH-1, ΔFgap1^σ^, and ΔFgap1^σ^-C strains after 2 weeks of inoculation on carrot agar plates. Bar = 50 μm.

**Figure 4 jof-09-00145-f004:**
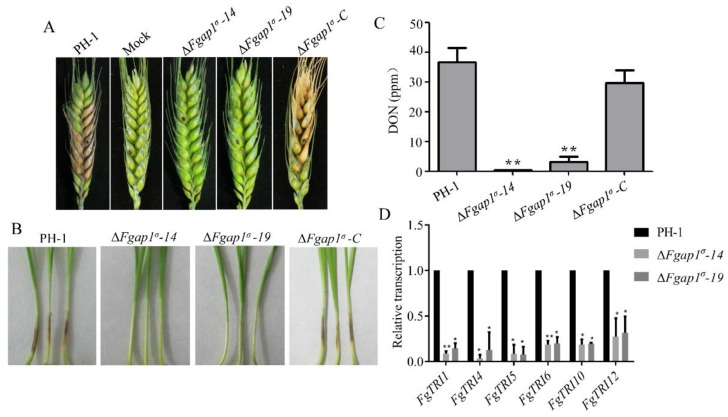
FgAP1^σ^ is required for virulence. (**A**) Pathogenicity of ΔFgap1^σ^ mutants on flowering wheat heads was significantly reduced compared with the wild-type PH-1 and ΔFgap1^σ^-C strains. Mycelia plugs from the various strains were inoculated on flowering wheat heads under moist conditions, and disease symptoms were recorded after two weeks. (**B**) The pathogenicity of ΔFgap1^σ^ mutants on wheat coleoptiles was abolished. (**C**) Level of deoxynivalenol (DON) production by ΔFgap1^σ^ mutants. Mycelia of the indicated strains were dried and weighed to quantify the fungal biomass. Level of significance was measured using an unpaired *t*-test (** *p* < 0.01). (**D**) The expression levels of some trichothecene biosynthetic genes FgTRI were significantly reduced in the ΔFgap1^σ^ mutants. Level of significance was measured using an unpaired *t*-test (* *p* < 0.05, ** *p* < 0.01).

**Figure 5 jof-09-00145-f005:**
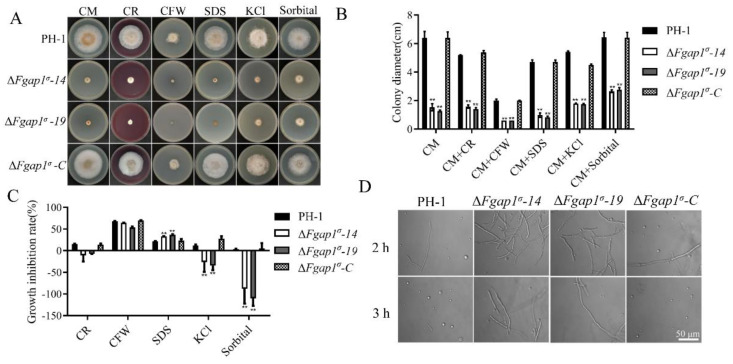
Deletion of FgAP1^σ^ resulted in cell wall integrity and osmotic pressure defects. (**A**) The PH-1, ΔFgap1^σ^, and ΔFgap1^σ^-C strains were incubated on CM plates supplemented with Congo red (CR, 200 μg/mL), calcofluor white (CFW, 200 μg/mL), SDS (0.01 %), sorbitol (1 M), and KCl (1 M) at 28 °C for 3 days. (**B**) Colony diameter of the indicated strains under cell wall and osmotic stresses. Level of significance was measured using an unpaired *t*-test (** *p* < 0.01). (**C**) Growth inhibition rate of the strains due to cell wall and osmotic stresses induced by the indicated agents. Level of significance was measured using an unpaired *t*-test (** *p* < 0.01). (**D**) Protoplast releasing potential of the PH-1 and ΔFgap1^σ^ mutants after treatment with cell-wall-degrading enzymes for 2 h and 3 h. Bar = 50 μm.

**Figure 6 jof-09-00145-f006:**
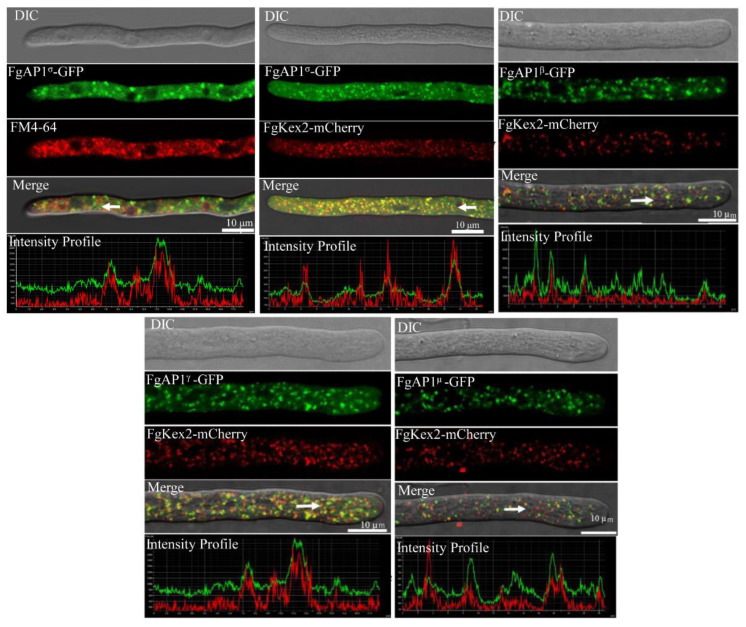
The localization of AP1 complex in *F. graminearum*. Colocalization of FgAP1^σ^-GFP with FM4-64 and trans-Golgi network (TGN, FgKex2-mCherry) as well as that of FgAP1^β^-GFP, FgAP1^γ^-GFP, and FgAP1^μ^-GFP with FgKex2-mCherry in *F. graminearum*. Images were captured from laser scanning confocal microscope. Bar = 10 μm.

**Figure 7 jof-09-00145-f007:**
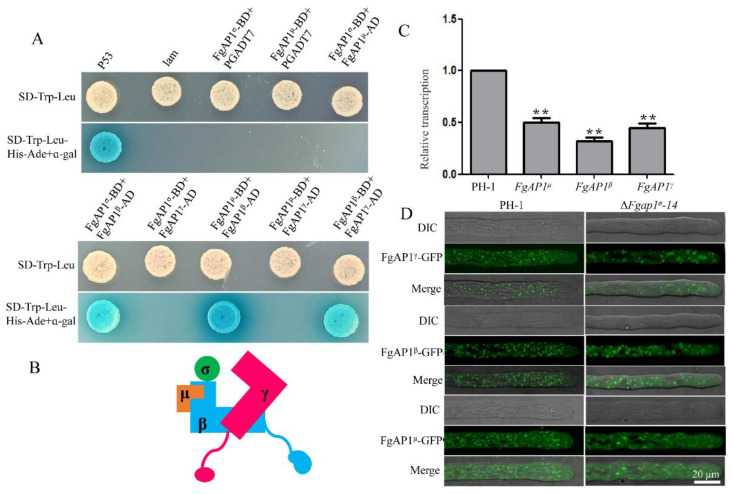
The relationship between FgAP1 complex subunits and the localizations of the various subunits in ΔFgap1^σ^ mutant. (**A**) Yeast two-hybrid assay showing the interactions of the AP1 complex subunits. There were positive interactions between FgAP1^β^ and FgAP1^σ^ and between FgAP1^γ^ and FgAP1^μ^. The interaction between pGBKT7-Lam and pGADT7-T was used as a negative control while that between pGBKT7-53 and pGADT7-T served as a positive control. (**B**) The interaction model of FgAP1^σ^ with the other subunits of FgAP1 complex. FgAP1^σ^ interacts with FgAP1^β^, and FgAP1^β^ interacts with FgAP1^γ^ and FgAP1^μ^, which indicates that FgAP1^σ^, FgAP1^β^, FgAP1^γ^, and FgAP1^μ^ function as a complex in *F. graminearum*. (**C**) The expression levels of FgAP1^β^, FgAP1^γ^, and FgAP1^μ^ were significantly downregulated in ΔFgap1^σ^ mutant. Level of significance was measured using an unpaired *t*-test (** *p* < 0.01). (**D**) The localizations of FgAP1^β^-GFP, FgAP1^γ^-GFP, and FgAP1^μ^-GFP were not affected in ΔFgap1^σ^ mutant. Bar = 20 μm.

**Figure 8 jof-09-00145-f008:**
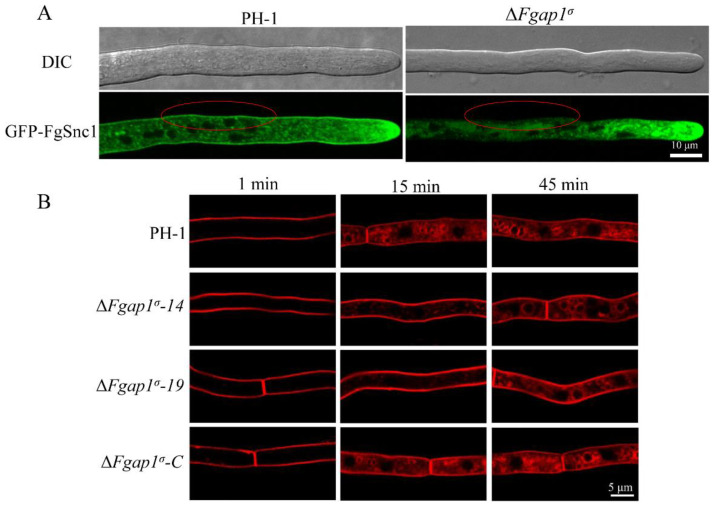
FgAP1^σ^ is involved in exocytosis and endocytosis processes. (**A**) The plasma membrane localization of GFP-FgSnc1 is disrupted in the ΔFgap1^σ^ mutant, as indicated by red circles. Bar = 10 μm. (**B**) FgAP1^σ^ delays internalization of FM4-64 into the vacuole membrane. Hyphae of the wild-type PH-1, ΔFgap1^σ^-14, ΔFgap1^σ^-19, and complement strain ΔFgap1^σ^-C were cultured in liquid CM for 24 h and then stained with FM4-64 and observed at different time points using a fluorescence confocal microscope. Bar = 5 μm.

## Data Availability

Not applicable.

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
