# Peer review of "FgAP1σ Is Critical for Vegetative Growth, Conidiation, Virulence, and DON Biosynthesis in Fusarium graminearum"

_jof, 2023, doi:10.3390/jof9020145_

Round 1
Reviewer 1 Report
The fusarium head blight disease is a critical global concern in wheat cultivation. in this context, the present manuscript is very valuable, informative and constructive from the perspective of resistance breeding in future. The methodology is appropriate and the content is written very well. I have a few queries and suggestions for the improvement of this manuscript
1. There is a need to include some sentences in the introduction as well as in the abstract regarding the futuristic implication of this work and how this finding is going to help in transgenic breeding and understanding of host-pathogen interaction.
2. Construction of fusion vector methodology needs to be elaborated for more clarity.
3. In the pathogenicity assay, how the inoculation was performed of the wheat kernels? what was the sample size? Are 7 days sufficient to observe the disease symptoms?
4. The germination of conidia in figure 3 D is not clearly distinguishable so please incorporate full-size images.
5. in figure 4A please specify on which day after inoculation the disease symptoms were recorded. it does not seem the 7-day-old disease development.
6. caption of Figure 7B (interaction model) requires a detailed explanation.
7. Figure 8A "The plasma membrane localization of GFP-FgSnc1 is abolished" not clear.
8. A concrete conclusion and future prospects may be added in the last section.
8.
Author Response
Comments and Suggestions for Authors
The fusarium head blight disease is a critical global concern in wheat cultivation. in this context, the present manuscript is very valuable, informative and constructive from the perspective of resistance breeding in future. The methodology is appropriate and the content is written very well. I have a few queries and suggestions for the improvement of this manuscript
- There is a need to include some sentences in the introduction as well as in the abstract regarding the futuristic implication of this work and how this finding is going to help in transgenic breeding and understanding of host-pathogen interaction.
Response:
We have added the sentences regarding the implication of this work and how the findings improve our understanding of host-pathogen interaction in the introduction and abstract.
- Construction of fusion vector methodology needs to be elaborated for more clarity.
Response:
This point is well taken. We have made the changes as suggested by the reviewer.
- In the pathogenicity assay, how the inoculation was performed of the wheat kernels? what was the sample size? Are 7 days sufficient to observe the disease symptoms?
Response:
For wheat spikelets infection assay, the wild type PH-1 and ΔFgap1σ strains were inoculated in liquid CMC medium and their conidia concentrations were adjusted to 4×104 cells/ml. Conidia suspensions were inoculated to fresh wheat coleoptile wounds and disease symptoms were observed 7 days after incubation at 25℃. Our results showed that deletion of FgAP1σ almost eliminated the pathogenicity in wheat kernels. We inoculated 7 wheat spikelets of each strain and the experiment was repeated three times. The wild type PH-1 and ΔFgap1σ-C strains produced serious disease symptoms after 7 days of inoculation. Previous researches for wheat spikelets infection assay were performed within 7 days [1-3] which was sufficient to observe the disease symptoms.
- The germination of conidia in figure 3 D is not clearly distinguishable so please incorporate full-size images.
Response:
We have made the changes as suggested by the reviewer.
- in figure 4A please specify on which day after inoculation the disease symptoms were recorded. it does not seem the 7-day-old disease development.
Response:
Mycelia plugs from the various strains were inoculated on flowering wheat heads under moist condition and the disease symptoms recorded after two weeks. We have added the specific days we recorded the disease symptoms as suggested by the reviewer in figure 4A.
- caption of Figure 7B (interaction model) requires a detailed explanation.
Response:
We have made the changes as suggested by the reviewer.
- Figure 8A "The plasma membrane localization of GFP-FgSnc1 is abolished" not clear.
Response:
The plasma membrane localization of GFP-FgSnc1 is disrupted in the ΔF-gap1σ mutant. We have replaced "abolished" with "disrupted" in Figure 8A.
- A concrete conclusion and future prospects may be added in the last section.
Response:
Many thanks to the reviewer for these helpful and constructive suggestions. We have added the concrete conclusion and future prospects in the last section.
- Zheng, Q.;Yu, Z.;Yuan, Y.;Sun, D.;Abubakar, YS.;Zhou, J.;Wang, Z.;Zheng, H The GTPase-Activating Protein FgGyp1 Is Important for Vegetative Growth, Conidiation, and Virulence and Negatively Regulates DON Biosynthesis in Fusarium graminearium. Front Microbiol. 2021, 12.
- Yang, C.;Li, J.;Chen, X.;Zhang, X.;Liao, D.;Yun, Y.;Zheng, W.;Abubakar, YS.;Li, G.;Wang, Z.; et al. FgVps9, a Rab5 GEF, Is Critical for DON Biosynthesis and Pathogenicity in Fusarium graminearum. Front Microbiol. 2020, 11.
- Yuan, Y.;Zhang, M.;Li, J.;Yang, C.;Abubakar, YS.;Chen, X.;Zheng, W.;Wang, Z.;Zheng, H.;Zhou, J The Small GTPase FgRab1 Plays Indispensable Roles in the Vegetative Growth, Vesicle Fusion, Autophagy and Pathogenicity of Fusarium graminearum. Int J Mol Sci. 2022, 23.
Reviewer 2 Report
Wu et al. performed experiments to study the biological functions of FgAP1σ, a subunit of the AP-1 complex in wheat pathogen F. graminearum. The results of this work expanded our understanding of AP-1 complex in disease causing fungi, and will help to find methods in controlling some crop devastating fungal diseases such as Fusarium head blight. However, several points should be concerned before the article could be accepted in publication.
1 the motivation of the work is just because of the the functional unclear of AP-1 complex in wheat pathogen Fusarium graminearum may greatly decrease the significances of the work.
2 those firstly appeared abbreviations should given the full definitions:
CFW and CR (line 23); SYM (line 110)
3 the big problem in the manuscript is the writing style in results and discussion sections., writings in these parts shows a very unprofessional way.
Results:
Result section included many contents which should be appeared only in discussion or introduction or even method parts. Some examples, “We searched for the predicted F. graminearum AP1 complex proteins using S. cerevisiae AP1 complex proteins (Aps1p, Apm1p, Apl2p, and Apl4p) [57] as a reference to perform a BLAST search against the available fungal genome, and identified….” (lines 189-191), should be moved to the method parts; “These results suggest that FgAP1σ is important for F. graminearum vegetative and polarized growth.” (lines 230,231), the results have clearly described above, I don’t think a conclusion herein is necessary; other similar points as “Asexual conidia and sexual ascospores produced by F. graminearum function as the important inocula that infect flowering wheat heads [58,59], In order to understand the function of FgAP1σ in conidiation, we inoculated the wild type PH-1, ΔFgap1σ-14, ΔF-gap1σ-19 and ΔFgap1σ-C strains in liquid carboxymethyl cellulose (CMC) media for 3 days to induce conidia production. (line 243-247), should be moved to the introduction or discussion part; “Taken together, these results suggest that FgAP1σ plays important roles in both asexual and sexual reproductions in F. graminearum. (line 259,260); as well as lines 281, 282; lines 283, 284; lines 294-296; lines 328-330; lines 344-347; lines 357, 358; line 367-370; lines 385, 386; lines 401-405; lines 412, 413; lines 422-424.
Discussion:
The discussion is descriptive and most of them just repeating the results. A deeper and comparative discussion between FgAP1σ functions in F. graminearum and in other microorganisms, or any innovation of the work will improve the value of the paper.
Author Response
Comments and Suggestions for Authors
Wu et al. performed experiments to study the biological functions of FgAP1σ, a subunit of the AP-1 complex in wheat pathogen F. graminearum. The results of this work expanded our understanding of AP-1 complex in disease causing fungi, and will help to find methods in controlling some crop devastating fungal diseases such as Fusarium head blight. However, several points should be concerned before the article could be accepted in publication.
1 the motivation of the work is just because of the the functional unclear of AP-1 complex in wheat pathogen Fusarium graminearum may greatly decrease the significances of the work.
Response:
The work is conceived as follows:Our previous studies revealed that the well-conserved FgAP2 complex subunits FgAP2β, FgAP2α, FgAP2mu and FgAP2σ are essential for polarized growth, development and pathogenicity of F. graminearum, and regulates the polar localization of the lipid flippases FgDnfA and FgDnfB [1]. In order to get a wider picture of the roles of FgAP complexes, we decided to make comparisons with the roles AP1 plays in the fungus. However, the functions of AP-1 complex in filamentous fungal species (including the devastating wheat pathogen F. graminearum) are still unclear. We therefore investigated the roles of FgAP1 complex to enable us systematically understand the function of FgAP complex in F. graminearum, and this would lay a solid foundation for effective prevention and control of Fusarium head blight (FHB). We have added the motivation of the work in the abstract and introduction parts.
2 those firstly appeared abbreviations should given the full definitions:
CFW and CR (line 23); SYM (line 110)
Response:
We have made the changes as suggested by the reviewer.
3 the big problem in the manuscript is the writing style in results and discussion sections., writings in these parts shows a very unprofessional way.
Results:
Result section included many contents which should be appeared only in discussion or introduction or even method parts. Some examples, “We searched for the predicted F. graminearum AP1 complex proteins using S. cerevisiae AP1 complex proteins (Aps1p, Apm1p, Apl2p, and Apl4p) [57] as a reference to perform a BLAST search against the available fungal genome, and identified….” (lines 189-191), should be moved to the method parts; “These results suggest that FgAP1σ is important for F. graminearum vegetative and polarized growth.” (lines 230,231), the results have clearly described above, I don’t think a conclusion herein is necessary; other similar points as “Asexual conidia and sexual ascospores produced by F. graminearum function as the important inocula that infect flowering wheat heads [58,59], In order to understand the function of FgAP1σ in conidiation, we inoculated the wild type PH-1, ΔFgap1σ-14, ΔF-gap1σ-19 and ΔFgap1σ-C strains in liquid carboxymethyl cellulose (CMC) media for 3 days to induce conidia production. (line 243-247), should be moved to the introduction or discussion part; “Taken together, these results suggest that FgAP1σ plays important roles in both asexual and sexual reproductions in F. graminearum. (line 259,260); as well as lines 281, 282; lines 283, 284; lines 294-296; lines 328-330; lines 344-347; lines 357, 358; line 367-370; lines 385, 386; lines 401-405; lines 412, 413; lines 422-424.
Response:
Many thanks to the reviewer for these helpful and constructive suggestions. We have removed some sentences, but we also retained some transitional and concluding sentences to make our article more complete.
Discussion:
The discussion is descriptive and most of them just repeating the results. A deeper and comparative discussion between FgAP1σ functions in F. graminearum and in other microorganisms, or any innovation of the work will improve the value of the paper.
Response:
Actually, the roles of AP1 complex in yeast, T. gondii, A. thaliana and mammals have been studied, however, studies on AP1 complex, especially with respect to its importance in growth and virulence, are limited in microorganisms, most notably in plant pathogenic fungi. This is the first report of the functions of AP-1 complex in the plant pathogenic fungi (Fusarium graminearum). We have discussed (and clearly elaborated) the novelty of our research work and compared the functions of FgAP1σ in F. graminearum and in other organisms such as yeast, T. gondii, A. thaliana and mammals in the revised manuscript.
- Zhang, J.;Yun, Y.;Lou, Y.;Abubakar, YS.;Guo, P.;Wang, S.;Li, C.;Feng, Y.;Adnan, M.;Zhou, J.; et al. FgAP-2 complex is essential for pathogenicity and polarised growth and regulates the apical localisation of membrane lipid flippases in Fusarium graminearum. Cell Microbiol. 2019, 21: 17.
Round 2
Reviewer 1 Report
All the suggestions are well incorporated.